# Integrating Chinese Herbs and Western Medicine for New Wound Dressings Through Handheld Electrospinning

**DOI:** 10.3390/biomedicines11082146

**Published:** 2023-07-30

**Authors:** Jianfeng Zhou, Liangzhe Wang, Wenjian Gong, Bo Wang, Deng-Guang Yu, Yuanjie Zhu

**Affiliations:** 1School of Materials & Chemistry, University of Shanghai for Science and Technology, Shanghai 200093, China; 221550217@st.usst.edu.cn (J.Z.); 223353279@st.usst.edu.cn (W.G.); 2Department of Dermatology, Naval Special Medical Center, Naval Medical University, Shanghai 200052, China; lzwang@hotmail.com (L.W.); m18817365409@163.com (B.W.)

**Keywords:** electrospinning, electrospraying, Yunnan Baiyao, ciprofloxacin, antimicrobial, combined medicine

## Abstract

In this nanotechnology era, nanostructures play a crucial role in the investigation of novel functional nanomaterials. Complex nanostructures and their corresponding fabrication techniques provide powerful tools for the development of high-performance functional materials. In this study, advanced micro-nanomanufacturing technologies and composite micro-nanostructures were applied to the development of a new type of pharmaceutical formulation, aiming to achieve rapid hemostasis, pain relief, and antimicrobial properties. Briefly, an approach combining a electrohydrodynamic atomization (EHDA) technique and reversed-phase solvent was employed to fabricate a novel beaded nanofiber structure (BNS), consisting of micrometer-sized particles distributed on a nanoscale fiber matrix. Firstly, Zein-loaded Yunnan Baiyao (YB) particles were prepared using the solution electrospraying process. Subsequently, these particles were suspended in a co-solvent solution containing ciprofloxacin (CIP) and hydrophilic polymer polyvinylpyrrolidone (PVP) and electrospun into hybrid structural microfibers using a handheld electrospinning device, forming the EHDA product E3. The fiber-beaded composite morphology of E3 was confirmed through scanning electron microscopy (SEM) images. Fourier-transform infrared (FTIR) spectroscopy and X-ray diffraction (XRD) analysis revealed the amorphous state of CIP in the BNS membrane due to the good compatibility between CIP and PVP. The rapid dissolution experiment revealed that E3 exhibits fast disintegration properties and promotes the dissolution of CIP. Moreover, in vitro drug release study demonstrated the complete release of CIP within 1 min. Antibacterial assays showed a significant reduction in the number of adhered bacteria on the BNS, indicating excellent antibacterial performance. Compared with the traditional YB powders consisting of Chinese herbs, the BNS showed a series of advantages for potential wound dressing. These advantages include an improved antibacterial effect, a sustained release of active ingredients from YB, and a convenient wound covering application, which were resulted from the integration of Chinese herbs and Western medicine. This study provides valuable insights for the development of novel multiscale functional micro-/nano-composite materials and pioneers the developments of new types of medicines from the combination of herbal medicines and Western medicines.

## 1. Introduction

Controlled drug release is a hot topic in the field of pharmaceutical technology. Conventional drug delivery systems, as the most widely used formulations, suffer from several drawbacks: large fluctuations in blood drug concentration, leading to the occurrence of “peak-trough” phenomena, where adverse effects may arise during peak drug concentration, while subtherapeutic drug levels during trough concentration can compromise therapeutic efficacy [1,2,3,4,5,6,7,8]. Hence, achieving fast or pulsatile drug release and sustained release can not only enhance treatment effectiveness but also reduce the frequency of administration, thus improving patient convenience [9,10,11,12,13,14]. 

Ciprofloxacin (CIP) is a synthetic third-generation fluoroquinolone antibacterial medication with broad-spectrum antibacterial action and excellent bactericidal characteristics. It has been reported to be many times more effective than norfloxacin and enoxacin. CIP also exhibits good antibacterial action and pharmacokinetic qualities with little adverse effects [15,16]. Traditional Chinese herbs are unique medicinal substances used in traditional Chinese medicine and serve as a significant hallmark distinguishing it from other medical practices. These herbs are abundant in resources and have a long history of use. Many of them contain natural active ingredients such as organic acids, flavonoids, terpenes, saponins, alkaloids, etc. These herbs exhibit antibacterial, antiviral, anti-inflammatory, antioxidant, anti-tumor, analgesic, immune-modulating, and tissue-regenerating activities, among others [17,18]. However, due to the standard pharmaceutical procedures utilized in the manufacturing of Chinese medications, there are obstacles in establishing controllable quality, safety, efficiency, and patient compliance in traditional Chinese medicine (TCM) [19]. Yunnan Baiyao (YB), a commercially available herbal preparation, has been used to treat bruises, injuries, and bleeding wounds. However, in practical use, the powdered form of YB is difficult to apply to the surface of wounds and cannot effectively eliminate bacteria around the wound. Therefore, it is possible to design a drug delivery system that combines CIP with YB to address these limitations.

In recent decades, the rapid advancement of nanotechnology and study of new excipients has led to significant improvements in the bioavailability of drugs, while simultaneously promoting safe, effective, and convenient drug delivery systems [20,21,22,23,24,25,26,27,28,29,30,31,32]. Hence, many newly proven safe and biocompatible excipients have been introduced, such as PVA, PVP, PEO, PCL, gelatin, etc. [33,34,35,36,37,38,39]. These excipients are designed to be combined with drug molecules to create nanomedicine products for disease treatment [40]. Traditional powders prepared as nanofiber membranes offer expanded dosage forms that are easy to administer and can combine both traditional Chinese and Western medicine to enhance therapeutic efficacy [41,42,43,44]. Incorporating traditional particles and capsules into electrospun nanofiber membranes facilitates drug delivery, improves taste, and enhances patient tolerance and treatment outcomes. The handheld electrospinning device is an emerging medical product that integrates electrospinning technology with existing medical techniques, providing functions such as immediate disinfection and wound management. Additionally, it is considered one of the smallest electrospinning apparatuses in the world, characterized by its compact size, light weight, and practicality. Driven by batteries, it allows for continuous production of nanofibers with a simple press of a button, enabling single-handed operation that seamlessly conforms to the user’s hand shape. It provides the flexibility to achieve in situ and directed deposition of various polymer nanofiber materials, marking a significant leap in the development of electrospinning devices.

In this study, electrospinning and electrospray coating were combined (both belonging to EHDA technique) to fabricate a novel hybrid material consisting of electrospun nanofibers and electrospray particles by using a handheld electrospinning device [45,46]. This material was developed for wound antimicrobial and healing applications. A series of characterization (SEM, XRD and FTIR), rapid disintegration, in vitro drug release, and antimicrobial study was conducted to evaluate the prepared hybrid nanofibers.

## 2. Materials and Methods

### 2.1. Materials

Yunnan Baiyao was purchased from a local Laobaixing Drugstore (Shanghai, China). Ciprofloxacin and ethanol were purchased from China National Pharmaceutical Group Corporation. Zein was bought from Shanghai Xingye Biological Technology Co., Ltd. (Shanghai, China). Polyvinylpyrrolidone (PVP K60) was purchased from Merck Chemical Reagent Co., Ltd. (Shanghai, China). Acetic acid and anhydrous ethanol were obtained from Shanghai Fitst reagent Factory (Shanghai, China). Milli-Q water was used throughout the study. All other chemical solvents were analytical grade.

### 2.2. Preparation of Precursor Solutions

Two different EHDA techniques were employed for the sequential preparation of Zein-YB microparticles via electrospray (single-fluid blending electrospraying) and CIP-PVP nanofibers via electrospinning (single-fluid blending electrospinning). Subsequently, a hybrid structure of nanofibers and microparticles, referred to as YB-CIP beaded nanofibers, was fabricated. Hence, three precursor solutions were prepared, corresponding to the electrostatic spray and electrostatic spinning processes. The specific parameters for the two EHDA techniques and the composition and concentrations of the three solutions are presented in Table 1.

### 2.3. Preparation of YB-CIP Beaded Fiber by Electrospinning

Firstly, Zein and YB were co-dissolved in a 100 mL solution of 75% ethanol. Microparticles, denoted as E1, were prepared using a simple electrospray process. Subsequently, 15 g E1 was uniformly dispersed into 200 mL of Fluid 2 with continuous stirring, forming a suspended working fluid denoted as Fluid 3. Then, the HHE-1 handheld electrospinning apparatus (Qingdao Junada Technology Co., Ltd., Qingdao, China) was used to fabricate the YB-CIP beaded fiber E3. The specific experimental conditions were as follows: the EHDA process is powered by 7th-grade Nanfu alkaline batteries (AAA), with a voltage range of 0~10 ± 1 kV (rated current generally below 90 mA) and a particle deposition distance of 20 cm. The ambient temperature and humidity were maintained at 21 ± 5 °C and 47 ± 7%, respectively.

### 2.4. Characterization

#### 2.4.1. Analysis of Morphology

The YB particles and nanofibers were used for morphology characterization by SEM (FEI Quanta G450 FEG, Inc., Hillsboro, OR, USA). Briefly, steps of fixing the prepared YB particles and nanofiber membrane to a sample holder with conductive adhesive, assuring a level surface under N_2_ environment, and subsequently applying a 1.5 min gold sputtering treatment using an ion sputter coater were conducted. A thin layer of 5 nm gold was deposited on the sample surface for SEM observation. ImageJ software v1.48 (National Institutes of Health, Bethesda, MD, USA) was used to measure the average diameter of the nanofibers.

#### 2.4.2. XRD and FTIR Analysis

X-ray diffraction (XRD) analysis of YB particles and nanofiber membrane was conducted using the D8 ADVANCE diffractometer (Bruker, Carteret, NJ, USA). The operating conditions included a working voltage of 40 kV and a tube current of 30 mA. The diffraction patterns were recorded in continuous mode within an angular range of 5° to 60° with a step size of 0.02° and a scanning speed of 5°/min.

FTIR spectroscopy of samples was recorded on a PerkinElmer FTIR Spectrometer (Spectrum 100, Billerica, MA, USA) by KBr method. The spectral curve was performed in the range 500–4000 cm^−1^ with a resolution of 2 cm^−1^.

### 2.5. Fast Dissolution Performance

The rapid disintegration and drug dissolution processes of the prepared drug-loaded nanofiber membrane E3 were evaluated by two self-developed methods. Briefly, one method included dropping a drop of water over a nanofiber mat collected on a glass slide, while the other method entailed depositing a piece of electrospun thin film on wet paper. All of procedures were photographed with a digital camera (Canon PowerShot SX50HS, Tokyo, Japan).

### 2.6. In Vitro Drug Release

The paddle method was used to assess the CIP release profiles of EHDA products in accordance with the Chinese Pharmacopoeia (2020 Edition). An amount of 25 mg EHDA product E2 and 0.1 g E3 were added to six vessels, which contained 500 mL phosphate-buffered solution (0.1 M, pH = 7.0). The dissolution media were maintained at 37 ± 1 °C with a rotation rate of 50 rpm. At a preset time point, 5.0 mL of the solution was withdrawn and filtered through a 0.22 μm film (Millipore, MA, USA). To keep the volume of the bulk solution constant, 5.0 mL of fresh PBS was added after sampling. A UV–vis spectrophotometer (UV-2102PC, Youke Instrument Co., Ltd., Shanghai, China) was used to measure the absorbance of CIP at λ_max_ = 276 nm. The experimental data were reported as mean ± standard deviation, and experiments were repeated 6 times.

There are multiple active ingredients in YB. The UV–vis spectrophotometer detection method cannot be exploited for quantitative analyses of YB sustained-release performance from the insoluble Zein microparticles. Thus, a direct observation with the eyes was conducted to assess the YB sustained release effect.

### 2.7. Antibacterial Performances of Nanofibers

The in vitro antibacterial effects of E1, E2, and E3 were evaluated using a plate count method. Gram-positive *Bacillus subtilis* (*Wb800*) and Gram-negative *Escherichia coli dh5α* (*E.coli dh5α*) were used as experimental microorganisms. The detailed process was as follows: (1) 5 mL of sterilized Luria–Bertani (LB) broth was loaded into an Erlenmeyer flask. (2) An amount of 50 mg EHDA products was placed into the LB broth solution, which contained about 1.5 × 10^5^ CFU of *E. coli dh5α* and *Wb800*, respectively. (3) The mixtures were incubated in a shaking incubator for 12 h at a constant temperature of 37 °C. (4) A total of 100 μL cell solution was seeded onto LB agar by surface spread plate technique. (5) After plates were incubated at 37 °C for 24 h, the number of CFU was counted.

Pure phosphate-buffered saline (PBS) was used as a blank control, and the antibacterial efficacy (ABE, %) of the specimen could be calculated according to the equation:ABE (%) = (Np − Nt)/Np × 100%
where Np and Nt represent the numbers of viable bacterial colonies of the blank control (pure PBS buffer added) and experimental group, respectively. All the experiments were performed six times, and the data were expressed as the mean values.

### 2.8. Statistical Analysis

All experiments were performed in triplicate throughout the study. The data were analyzed and plotted using Origin Pro 2021 (Origin Lab Co., Northampton, MA, USA). 

## 3. Results and discussion

### 3.1. The Sequential EHDA Process

Electrohydrodynamic atomization (EHDA) utilizes electrical energy to directly evaporate solvents, rapidly drying and solidifying microfluids within a timescale of 10^−2^ s, resulting in the generation of materials at the micro/nanoscale [47,48]. EHDA mainly comprises two processes: electrospinning and electrospray. The principles of electrospinning and electrospray are similar, with two differences. Firstly, it lies in the concentration of the polymer solution. When the liquid concentration is high, the jet from the Taylor cone is relatively stable, leading to fiber deposition on the receiving device [49]. Conversely, when the liquid concentration is low, the jet from the Taylor cone becomes unstable due to the effect of high electric voltage, resulting in the transformation of larger droplets into smaller ones and ultimately forming particles. Secondly, it depends on the degree of interaction between polymer solutions. If the degree of interaction is high, fibers are formed; if the degree of interaction is low, particles are formed. Therefore, by adjusting the concentration of the spraying solution and the type of polymer, the transition between electrospinning and electrospray can be achieved.

The working fluid for an effective EHDA technique has to be electrospinnable [50]. It can be understood that electrospinning typically produces solid products in the form of nanofibers through the continuous stretching of a viscous polymer fluid jet [51,52]. On the other hand, electrospray coating typically generates solid products in the form of particles through droplet fragmentation and repulsion [53]. Hence, A new type of hybrid material on different scales can be envisioned based on the outcomes of these two EHDA processes, as shown in Figure 1.

Electrospraying possesses advantages such as simple preparation, good reproducibility, high drug loading capacity, controllable particle size and surface morphology, and narrow size distribution. Moreover, during the spraying process, the droplets carry like charges, resulting in mutual repulsion and excellent self-dispersion. This property prevents particle aggregation and enables the production of monodisperse drug-loaded nanoparticles. Therefore, it has been widely applied in the field of pharmaceuticals.

The electrospraying process was performed using an electrospinning device. The working Fluid 1 was shown in Figure 2a. As shown in Figure 2b, the entire electrospraying process was successfully carried out, and product E1 was collected on the collector, exhibiting a white area. Figure 2c is an enlarged view of the needle during the electrospraying process, where the Taylor cone is manifested as a typical cone shape. The droplets are affected by Coulomb repulsion and are fragmented into tiny droplets in the air, forming particles that scatter and deposit on the collector as the solvent rapidly evaporates.

The electrospinning process was performed by HHE-1 handheld electrospinning device. Working Fluid 3, depicted in Figure 3a, had a brown color. It was loaded into a 5 mL syringe, connected to a stainless-steel needle, and inserted into the handheld device along with a stainless-steel needle. As shown in Figure 3b, as the device button was continuously pressed, electrospun fibers (E3) were collected on the collector (white area in Figure 3b). In the magnified view of Figure 3b, the fiber-beaded surface morphology can be clearly observed. The liquid flow rate from the stainless-steel needle was carefully monitored, and the electrospinning effect was observed. The electrospinning process was shown in Figure 3c, where a straight jet is ejected from the Taylor cone at the needle tip, which gradually bends and undergoes whipping phenomena as the solvent evaporates, eventually forming fibers [54].

### 3.2. Morphology of Product

The morphological changes of different EHDA products are shown in Figure 4. Irregularly arranged YB–Zein particles were obtained on aluminum foil, exhibiting excellent dispersion without significant agglomeration (Figure 4a). Interestingly, there were hardly any “satellites” (submicron particles) observed around the electrosprayed YB–Zein particles (Figure 4a). Due to their small size, these “satellites” may be regarded as a negative aspect for drug release applications because the loaded drug molecules have a short diffusion path to reach the surrounding fluid. As expected, the electrospun CIP-PVP nanofibers (E2) exhibited fine linear morphology. The fibers showed no adhesion, interweaving with each other, and had a certain porosity (Figure 4b). Additionally, the diameter of the E2 electrospun fibers was about 640 ± 130 nm (Figure 4e). PVP is a linear polymer known for its good spinnability. Moreover, PVP has excellent solubility in water, ethanol, methanol, acetone, and other organic solvents [55,56]. Hence, the FDA has approved PVP for biomedical uses, and it is frequently investigated for various dosage forms such as powders, particles, tablets, and capsules. Furthermore, Figure 4b revealed a smooth surface of E2 nanofibers with no drug particles formed due to phase separation during the static process. The electrospun hybrid material YB-CIP beaded nanofiber (E3) prepared from the suspension working Fluid 3 exhibited a mixture of typical beads or spindle bodies along with nanofibers, as shown in Figure 4c,d.

The size of the EHDA products was evaluated using ImageJ software. The average diameter of CIP-PVP nanofibers (E2) and YB particles (E1) was determined to be 640 ± 130 nm (Figure 4e) and 2.56 ± 0.78 μm (Figure 4f), respectively. The diameter of the PVP fibers and Zein particles in the YB-CIP beaded nanofiber (E3) was measured to be 610 ± 320 nm (Figure 4g) and 2.85 ± 0.79 μm (Figure 4h), respectively. The difference in PVP fiber diameter between E2 and E3 was attributed to the addition of Zein particles. The E1 particles were uniformly distributed within the fibers, and their addition increased the stretching effect on the flowing fluid jet, resulting in an uneven thickness. However, there was no significant difference in the average diameter of Zein particles between E1 and E3, indicating that E1 could be redispersed as a suspension in Working Fluid 3 without altering the particle size.

### 3.3. Physical State and Compatibility of Components

The crystalline nature of raw materials and compounds were analyzed by X-ray diffraction. As shown in Figure 5, characteristic peaks corresponding to CIP and YB are observed in their respective XRD spectra, indicating the crystalline form of CIP and YB. However, the XRD spectra of PVP, Zein, and hybrid compounds exhibit broad peaks within specific angular ranges, indicating the presence of amorphous phases. The broad peaks suggest a lower degree of crystallinity in the samples. PVP is an amorphous linear polymer known for its excellent electrospinnability, which prevents the crystallization of many drugs. It can be inferred that the crystalline component of CIP loses its crystal state during the electrospinning process and exists in an amorphous form within the hybrid membrane. This result is favorable for the rapid dissolution and release of the active ingredient CIP, as it eliminates the need to overcome lattice energy for dissolution [57,58].

The FTIR spectra of the materials (CIP, YB, Zein, and PVP) and hybrid were shown in Figure 6a. The characteristic peak at 3275 cm^−1^ corresponds to the stretching vibration of O-H in CIP. The characteristic peak at 3054 cm^−1^ is attributed to the stretching vibration of C-H, and the absorption peak at 1986 cm^−1^ represents the vibration of water molecules absorbed by CIP. Additionally, there are several sharp peaks in the fingerprint region of CIP. However, when CIP is encapsulated in compounds containing corn protein, these sharp peaks almost disappear. It could be explained by secondary interactions between the (O-H) and (C-H) of CIP molecules and the Zein carriers (Figure 6b), which typically include hydrogen bonding, hydrophobic contacts, and electrostatic interactions [59]. On the one hand, this facilitates the rapid dissolution of CIP. On the other side, it contributes to the stable transport and storage of the thin film. Moreover, in the hybrid spectrum, the characteristic peak of C=O was found at 1658 cm^−1^, which exhibits a slight blue shift compared to Zein (1661 cm^−1^) and a slight red shift compared to PVP (1658 cm^−1^). Combined with the molecular structure in Figure 6b, this further confirms the interactions between the components and the encapsulation effect of E3 on CIP and YB.

### 3.4. Rapid Disintegration of YB-CIP Beaded Nanofiber Membrane

The dissolution experiment was conducted using a glass slide, as shown in Figure 7a. After depositing YB-CIP fibers for 5 min, a drop of water was added to the YB-CIP fibers. A camera was used to photograph the rapid dissolution process. Upon contact with the water droplet, the YB-CIP fibers rapidly disintegrated into a transparent gel, gradually revealing the logo of “USST” The time from “a1” to “a6” was about 4.8 ± 0.4 s. By morphological analysis of the water-dripped place, different images were observed. One typical image is that the water-soluble CIP-PVP nanofibers were swollen and dissolved upon water absorption, while the water-insoluble Zein particles dispersed uniformly on the glass slide (Figure 7b). Another typical image is that the dissolved drug CIP precipitated and re-crystallized into elongated strips, as shown in Figure 7c.

Furthermore, as shown in Figure 8a, the fiber membrane was cut into circular pieces with a diameter of 1.2 cm using a puncher, which was used to simulate the rapid disintegration experiment of an artificial tongue. First, the water-soaked filter paper was placed in a Petri dish, and then a circular YB-CIP fiber membrane was placed on the surface of the wet paper. As shown in Figure 8b, the dissolution and passive diffusion process of the YB-CIP fiber membrane transformed into a water-absorbing gelation process, which was captured by a camera. The time from “1” to “6” was about 1.7 ± 0.4 s. The fiber membrane changed from opaque white to semi-transparent. Obviously, the gelation process was promoted by the hygroscopicity of the PVP matrix, the tiny diameter of the nanofibers, and the three-dimensional network structure of the nanofiber membrane. It is worth noting that a faint yellowish color gradually shows in fiber membranes “7” to “8” in Figure 8b. This process had a relatively long duration of approximately 91.7 ± 11.4 s, indicating a passive diffusion process. However, the YB–Zein particles could not be dissolved and removed; thus, a light-yellow trace was left there, as indicated by the red arrow in Figure 8b-8. Of course, when additional stirring was introduced (e.g., mimicking tongue movements or a wound place), the diffusion and transportation process remained very rapid. It should be mentioned that due to the hygroscopic nature of the polymer carrier (i.e., PVP), the membrane needs to be stored in a low-humidity environment, which is a common concern for many conventional dosage forms.

### 3.5. In Vitro Drug Release

The in vitro release profiles of CIP from E2 and E3 were determined using UV–vis spectrophotometry. The wavelength scan of CIP in the range of 170–370 nm was performed, as shown in Figure 9a. The results indicated maximum absorbance of CIP at 276 nm. Subsequently, a series of CIP solutions with different concentrations was formulated to construct a standard curve of *A* = 0.1148 *C* + 9.8924 × 10^−4^ with a R value of 0.9997 (Figure 9b), where *A* was the absorbance and *C* was CIP concentration (1–50 μg/mL).

The CIP release performances of EHDA product E2 and E3 are shown in Figure 9c,d, respectively. These release profiles were constructed based on the accumulative percentage of CIP released over time. Clearly, both the CIP-PVP nanofibers E2 and the beads-on-a-string E3 exhibited a pulsatile release of the loaded CIP within 1 min. These results demonstrated that the E3, on one hand, was able to release the Western medicine as rapidly as the electrospun nanofibers E2, by which it can be expected that the therapeutic effect of CIP can be rapidly initiated for urgent wound treatment requiring fast antimicrobial activity. On the other hand, E3 contained the insoluble YB-Zein microparticles, which could manipulate the sustained release of active ingredients in YB for playing their important roles in wound healing (Figure 9e). YB is famous for its excellent functions such as promoting myogenesis and stopping pain and detumescence. A sustained release profile of YB is better for this functional performance.

It is noteworthy that rapid release of CIP can be attributed to two main factors: (1) the excellent hydrophilic nature of the PVP K60 matrix [55] and (2) the electrospun membrane exhibits characteristics such as a large fiber surface area, small diameter, and high porosity [60,61,62], which accelerate the dissolution and diffusion of CIP from the PVP matrix. In contrast, the YB released from the Zein particles through a diffusion process due to the insolubility of Zein in water. The micro-scale diameters mean a longer way for the YB diffusion routes and thus a better sustained release effect. Although UV–vis cannot be explored for quantitative analyses of the YB release profiles due to multiple active ingredients, the direct observations of the taupe color by eyes can judge the sustained release effect of YB to 12 h.

Additionally, an interesting phenomenon is the color change of the materials conversion. The raw YB powders show a gray color. In Figure 2a, the YB–Zein solution showed a taupe color, and the YB–Zein particles E1 showed a deeper taupe color. However, the suspensions for handheld electrospinning showed a yellow color in Figure 3a, and the electrospun E3 showed almost a white color (Figure 8a). In the YB sustained-release experiments, the taupe colors were gradually increased as the YB ingredients were gradually freed from the Zein particles to the dissolution media in Figure 9e.

### 3.6. Analysis of Antibacterial Performances

As a useful wound dressing, the first and foremost thing is its antibacterial performance [63,64,65]. Bacterial counts and antibacterial rates in the culture medium were determined using the plate counting method at 6, 12, and 24 h of incubation. The *Escherichia coli* and *Staphylococcus aureus* represent Gram-negative and Gram-positive bacteria, respectively. As shown in Table 2, the powders of YB have a certain antibacterial performance. When they were loaded into Zein microparticles through electrospraying, the formed EHDA product E1 still had antibacterial effects. Furthermore, with the increase of incubation time, the antibacterial effects increased significantly, suggesting the sustained release of active ingredients from the Zein particles. For *Wb800* and *Escherichia coli dh5α*, the increases are from 88.3% to 99.9% and from 81.1% to 99.9% after 2 and 12 h, respectively.

As for the little molecule CIP, it has a fine sterilizing effect. After 2 h of incubation, the ABE% could reach a value of around 99.0%, suggesting its fast effect on antibacterial performance. As anticipated, when the YB-loaded microparticles were combined with the electrospun CIP-loaded PVP nanofibers, synergistic antibacterial effects were achieved, having fast initiation and also an extension effect in antibacterial performance. YB is famous for its hemostatic and myogenic effects. Its encapsulation into Zein particles can benefit the myogenic effects for wound healing, whereas the combination with CIP can endow the hybrid EHDA products with a desired antibacterial performance. The strategy of tailoring components, compositions, and organization formats through advanced fabrication methods can promote the development of novel biomedicines [66,67,68]. The present protocol is a fine example of taking into consideration several factors together to create new kinds of medicated products. Additionally, the handheld electrospinning in situ for treating open wound places should be more convenient than the raw powders of YB.

### 3.7. Release Mechanism

The process of multi-drug release from the hybrid product E3, which is made up of electrospun PVP nanofibers and electrosprayed YB–Zein particles, is rather evident based on the analysis outlined above, as shown in Figure 10. When the hybrid product E3 is immersed in the dissolution medium, the CIP-PVP nanofibers dissolve rapidly. This represents a typical first-stage erosion process for the rapid release of CIP. Then, YB molecules distributed or adsorbed on the surface of YB–Zein particles dissolve into the dissolution medium, gradually opening pathways for water molecules to penetrate the interior cross-sections of the Zein particles. As water molecules permeate from the surface to the core of Zein particles, the loaded YB molecules freely diffuse back into the bulk solution from the Zein particles. Throughout the entire process, the Zein skeletons maintain the diffusion and exchange of water and YB molecules. In theory, the diffusion process continues until achieving a uniform drug concentration distribution throughout the entire bulk solution and reaching dynamic equilibrium in terms of absorbance between the dissolution medium and the solid Zein skeletons. Human health always relies on the establishment of new methods [69,70,71,72], with the continuous emergence of “bottom-up” approaches involving molecular reactions to create new materials and “top-down” methods for preparing nanomaterials [73,74,75]. In this study, the trans-scale combination of nanofibers and microparticles has been utilized to generate functional hybrid materials, enhancing the functional performance of the materials. Meanwhile, the release profile of a drug always relies on the property of the host matrix that is exploited to encapsulate the drug [76,77,78,79]. Here, another combination of water-soluble PVP and water-insoluble Zein has been utilized to provide the different release behaviors of different drugs. The above-mentioned two “combinations” are finally aimed to the third combination, i.e., the combination of Chinese Herbs and Western Medicine for an improved wound healing effect.

## 4. Conclusions

In this study, a continuous EHDA process was developed to prepare a novel drug mixture, E3. Zein was used as a sustained-release carrier PVP K60 was used as the spinning matrix; and YB and CIP were used as model drugs for hemostasis, analgesia, and antimicrobial purposes, respectively. A handheld electrospinning device was successfully employed to fabricate bead-on-string structured nanofiber blend membranes for rapid antimicrobial activity and long-lasting hemostasis. SEM revealed well-dispersed YB–Zein particles with almost no unfavorable smaller particles on the surface, providing the possibility of sustained release of YB. The CIP–PVP fibers exhibited a good linear morphology with a uniform and smooth surface, free from beads or spindles, enabling rapid drug release for antimicrobial efficacy. The YB–CIP hybrid membrane presented a fiber-beaded structure. In vitro dissolution tests demonstrated the complete release of the antimicrobial drug CIP within 1 min. The prepared E3 exhibited desirable functional performance in facilitating the rapid breakdown and dissolution of CIP, thereby enhancing patient convenience. The antimicrobial analysis experiment demonstrated the excellent antibacterial performance of YB–CIP against *Escherichia coli*. Based on the combination of handheld electrospinning and electrospraying and the combination of nanofibers of soluble polymeric matrix and microparticles of insoluble protein, a new concept about the development of combined medicine from the traditional Chinese herbs and modern Western medicine was demonstrated. The present protocols pioneered a new approach for developing many new sorts of combined biomedicines.

## Figures and Tables

**Figure 1 biomedicines-11-02146-f001:**
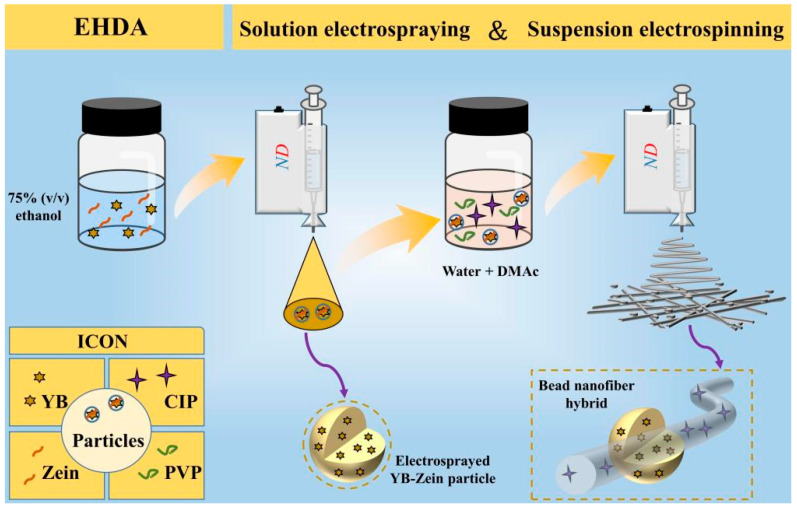
A schematic diagram of the fabrication for bead-on-string nanofibers composed of electrospun nanofibers and electrospray particles.

**Figure 2 biomedicines-11-02146-f002:**
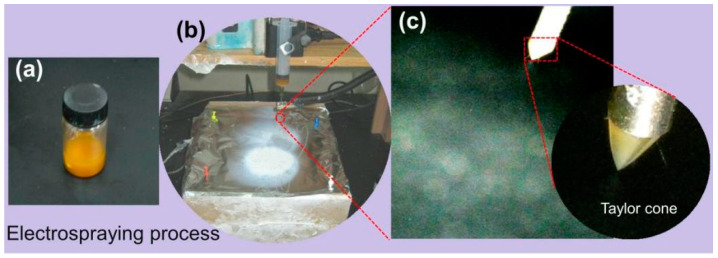
The typical electrospraying process: (**a**) The working Fluid 1; (**b**) Electrospraying equipment operation; (**c**) The electrospraying process and Taylor cone.

**Figure 3 biomedicines-11-02146-f003:**
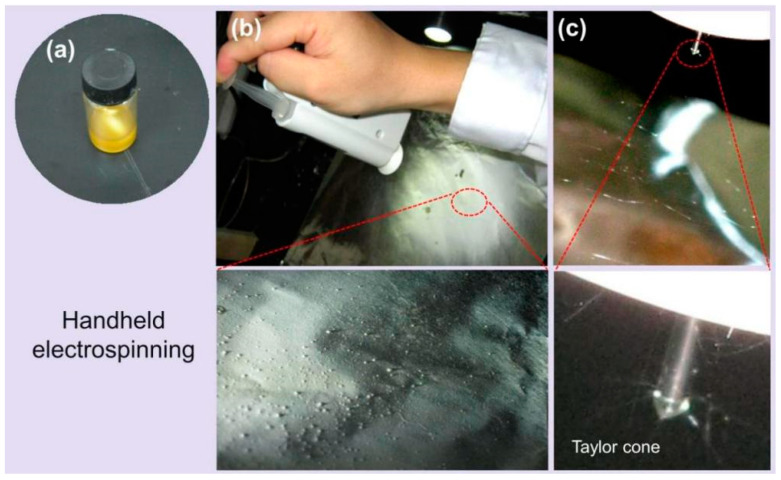
Hand-held electrospinning equipment and typical electrospinning process: (**a**) The working Fluid 3; (**b**) Hand-held electrospinning equipment operation; (**c**) The electrospinning process and Taylor cone.

**Figure 4 biomedicines-11-02146-f004:**
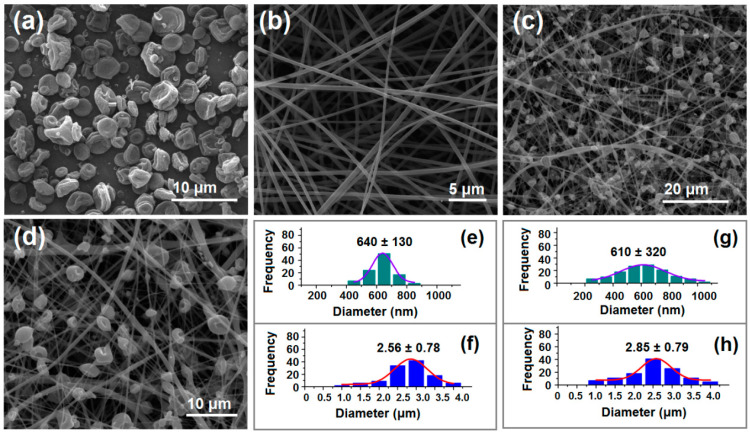
SEM images and the average diameters of EHDA products. (**a**) YB–Zein particles E1; (**b**) CIP-PVP nanofibers E2; (**c**) Hybrid YB-CIP nanofibers E3; (**d**) Magnified image of the hybrid YB-CIP nanofibers E3; (**e**) Average diameter of CIP-PVP nanofibers E2; (**f**) Average diameter of YB–Zein particles E1; (**g**) Average diameter of CIP-PVP nanofibers E3; (**h**) Average diameter of hybrids E3 microparticles.

**Figure 5 biomedicines-11-02146-f005:**
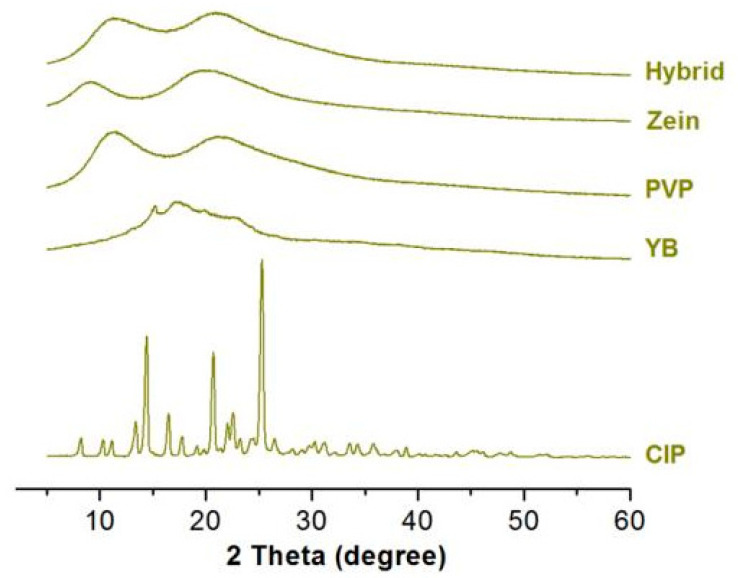
XRD spectra of CIP, YB, PVP, Zein, and Hybrid.

**Figure 6 biomedicines-11-02146-f006:**
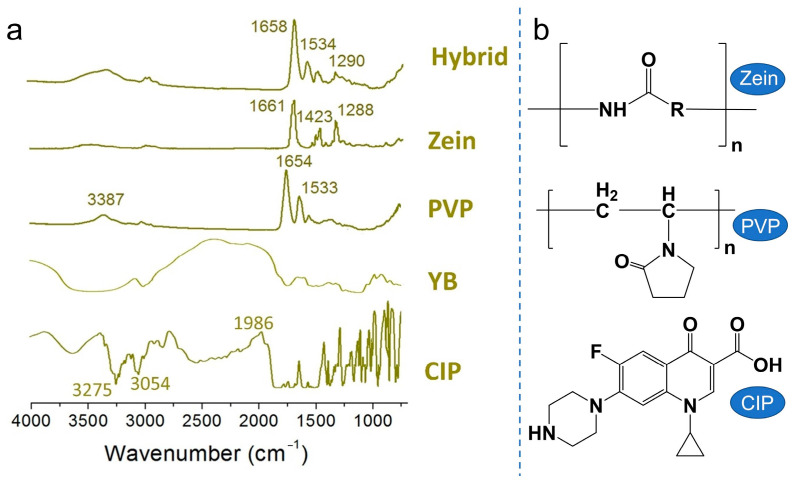
(**a**) FTIR spectra of CIP, YB, PVP, Zein, and Hybrid; (**b**) The molecular formats of the components (Zein, PVP, and CIP).

**Figure 7 biomedicines-11-02146-f007:**
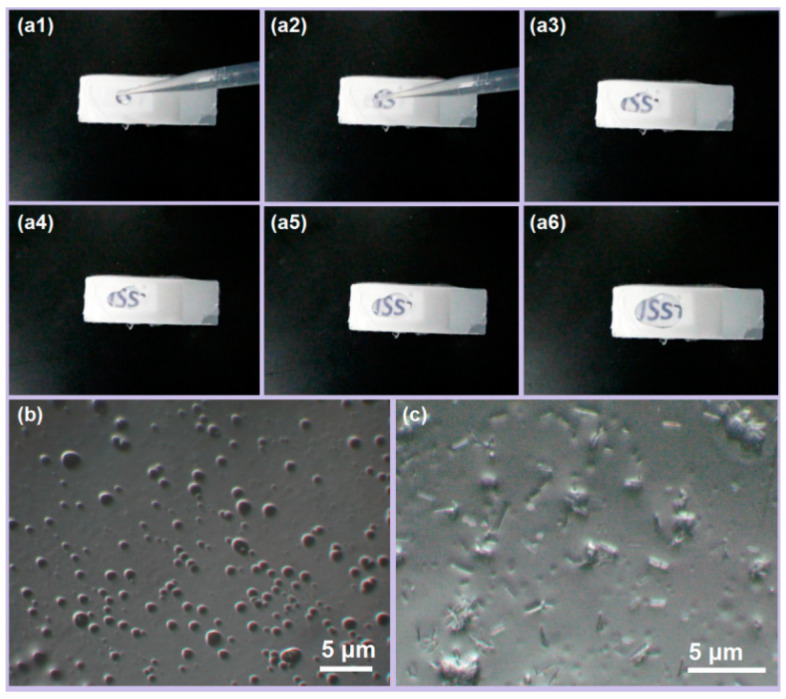
Rapid disintegration of YB-CIP beaded nanofiber membrane. (**a1**–**a6**) Dissolution process; (**b**,**c**) Different OM images of the water-dripped place after drying.

**Figure 8 biomedicines-11-02146-f008:**
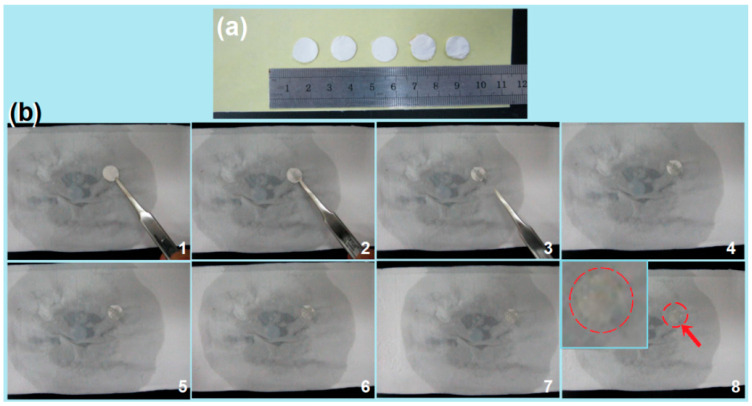
Rapid disintegration experiment of simulated artificial tongue. (**a**) Circular YB-CIP fiber membrane; (**b**) Artificial tongue process, the dissolution process is according to the order from “1” to “8”, the up-left inset of “8” is an enlarged image indicted by the red arrow and circle, in which are YB-Zein particles.

**Figure 9 biomedicines-11-02146-f009:**
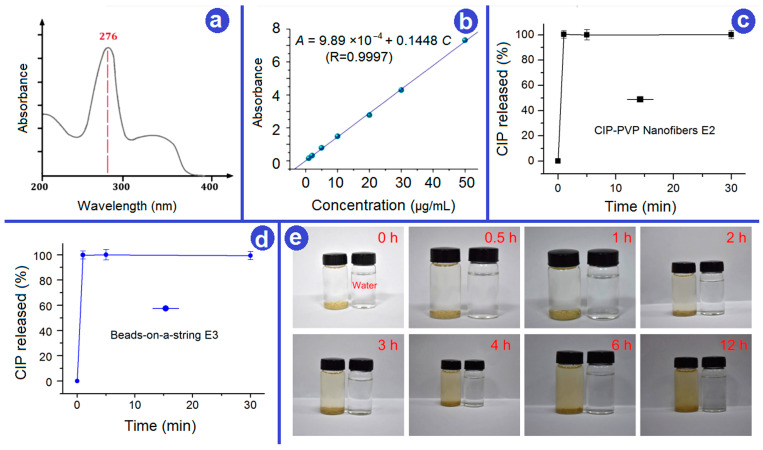
In vitro drug release. (**a**) Absorbance curve of CIP; (**b**) Standard calibration curve of CIP; (**c**,**d**) In vitro CIP pulsatile release from E2 and E3, respectively; (**e**) An observation of the YB sustained release effect from the electrosprayed insoluble Zein microparticles E1.

**Figure 10 biomedicines-11-02146-f010:**
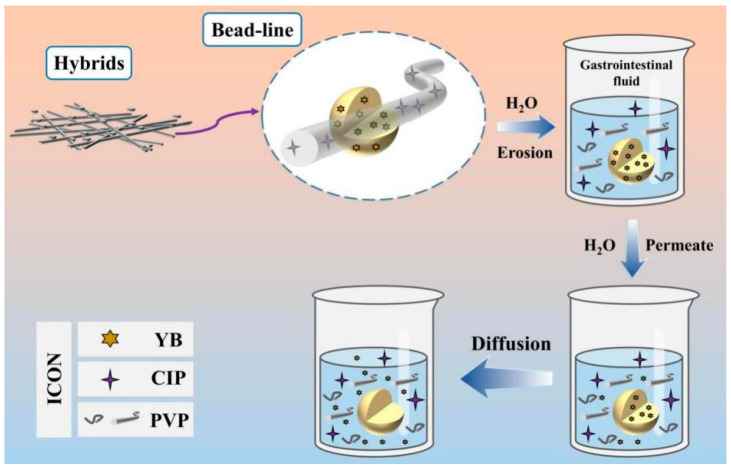
Multi-drug dissolution-controlled release and synergistic mechanism.

**Table 1 biomedicines-11-02146-t001:** Parameters for the EHDA processes.

No.	EHDA Process	Working Fluid	Experimental Conditions		Drug Contents	Morphology
			V (kV)	D (cm)		
E1	Electrospraying	Fluid 1 ^a^	20	20	20.0% YB	Particles
E2	Electrospinning	Fluid 2 ^b^	Cell	20	20.0% CIP	Fibers
E3	Sequential EHDA process	Fluid 3 ^c^	Cell	20	15% (YB) & 5% (CIP)	Hybrids

^a^ **Fluid 1**: An amount of 3.0 g YB and 12.0 g Zein were co-dissolved in 100 mL 75% ethanol solution. ^b^ **Fluid 2**: An amount of 8.0 g PVP and 2.0 g CIP were co-dissolved into 100 mL mixture of ethanol and acetic acid with a volume ratio of 90:10. ^c^ **Fluid 3**: An amount of 15.0 g microparticles E1 from electrospraying were suspended into 50 mL of Fluid 2 uniformly through continuous stirring.

**Table 2 biomedicines-11-02146-t002:** The antibacterial performance of the EHDA products against *Wb800* and *Escherichia coli dh5α* (*n* = 6) ^a^.

Bacteria	Samples	Initial CFU	CFU after 2 h	CFU after 6 h	CFU after 12 h
			CFU (ABE%)	CFU (ABE%)	CFU (ABE%)
*Wb800*	YB	1.5 × 10^5^	2.8 × 10^4^ (88.3%)	1.4 × 10^4^ (98.7%)	6.7 × 10^3^ (99.9%)
	E1	1.5 × 10^5^	4.7 × 10^4^ (80.4%)	2.4 × 10^4^ (97.8%)	8.5 × 10^3^ (99.9%)
	E2	1.5 × 10^5^	1.2 × 10^3^ (99.5%)	2.8 × 10^2^ (>99.9%)	1.9 × 10^2^ (>99.9%)
	E3	1.5 × 10^5^	2.1 × 10^4^ (91.3%)	3.6 × 10^2^ (>99.9%)	2.1 × 10^2^ (>99.9%)
	Blank	1.5 × 10^5^	2.4 × 10^5^	1.1 × 10^6^	7.3 × 10^6^
*Escherichia coli dh5α*	YB	1.5 × 10^5^	5.1 × 10^4^ (81.1%)	2.2 × 10^4^ (98.1%)	4.7 × 10^3^ (99.9%)
	E1	1.5 × 10^5^	7.4 × 10^4^ (72.6%)	3.5 × 10^4^ (97.1%)	5.3 × 10^3^ (99.9%)
	E2	1.5 × 10^5^	4.2 × 10^3^ (98.4%)	8.9 × 10^2^ (>99.9%)	3.4 × 10^2^ (>99.9%)
	E3	1.5 × 10^5^	7.6 × 10^3^ (97.2%)	1.3 × 10^3^ (>99.9%)	5.7 × 10^2^ (>99.9%)
	Blank	1.5 × 10^5^	2.7 × 10^5^	1.2 × 10^6^	8.1 × 10^6^

^a^ Abbreviations: YB, Yunnan Baiyao powders (10 mg, an equal amount loaded in microparticles E1); CFU, colony-forming units; ABE, antibacterial efficacy.

## Data Availability

The data supporting the findings of this manuscript are available from the corresponding authors upon reasonable.

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
