# Peer review of "Integrating Chinese Herbs and Western Medicine for New Wound Dressings through Handheld Electrospinning"

_biomedicines, 2023, doi:10.3390/biomedicines11082146_

Round 1

Reviewer 1 Report

Elaboration and implementation of nanotherapeutic platforms to supply the controlled drug delivery with a desired kinetic profile are critical issues for innovative biomedicine. The authors suggest novel biopolymer pharmaceutical systems produced via electrospinning/electrospraying nanotechnologies. 

The positive item of the manuscript includes the usage of up-to-date techniques to explore the structural and antibacterial behavior of innovative therapeutic systems as well as the EHDA method. The experiment details cofirm the presentation's principal topics, except the “West medicine – China treatment” interpenetration lack.   

With the amendments of manuscript content and the improved argumentation, the manuscript’s structure has a coherent and reasonable text presentation with good-quality of the illustrations. The literature is presented rationally.  

The authors are invited to check the typos/small grammatical flaws and to correct the remark.               

The idea of the combination between specific Chinese herb therapy and conventional west-spread medicine, which was announced in the Title, remains practically undisclosed and should be developed  in the submission, specifically in the Abstract and in the Conclusions section. 

Fig. 9C: Because of the extremely high rate of CIP delivery, the authors did not be able to receive the drug release profile, therefore this part of the graph should be taken fairly tentatively.   

In the legend to this Figure, it is worth indicating the kind of system loaded by the drug 

P4, L174: within a timescale of 10-2 s  it is rather a typo – please amend

To summarize, the content of the submission falls into the scope of “the Biomedicine” journal, and it is worth recommending this paper for the following Edition/Publishing performance after making the above amendments.

The Quality of scientific English is quite appropriate.

Author Response

See attached file please!

Reviewer 2 Report

Although the paper is very interesting, authors should provide FTIR spectra of fabricated fibers. There was no FTIR data represented on the nanofibers. 

Authors should also provide the zeta potential of electrospinning liquids to understand the ionic interactions of the final solutions before electrospinning process.

Author Response

See attached file please!

Reviewer 3 Report

The analyzed article is of value, well prepared and realized. The experiments are properly conducted, and the interpretation of the results supports the experimental data very well.

1. As a particularly important observation that I want to make, it is related to the remedy Yunnan Baiyao (YB). The composition of these Chinese plants and the elements that are responsible for the therapeutic applications are protected by elements of industrial intellectual property. However, it is important to report some more details regarding the use of this remedy, a series of adverse reactions described. If the composition is standard, and if it is part of the category of herbal therapeutic supplements, what is the regime of these products?

2. The graphic part of the article manuscript is correctly rendered, but the resolution of the figures should be standardized according to the requirements.

3. The bibliographic references are appropriate. Compliance with the instructions in the Guide for authors, at the end, must be verified.

Based on your own proficiency in English, is required moderate editing of English language on the manuscript. 

Author Response

See attached file please!

Reviewer 4 Report

In this manuscript, the authors developed a novel beaded nanofiber formulation E3 with both Yunnan Baiyao loaded Zein nanoparticles and ciprofloxacin PVP fiber via EHDA technique and reversed-phase solvent. The morphology of E3 was characterized by SEM. Amorphous state of CIP in the E3 formulation was confirmed by XRD and FTIR. The in vitro release of CIP and antibacterial efficacy were evaluated. However, there are some concerns that need to be addressed.

1. Please correct the ‘5’ of ‘105’ in superscript form on line 156

2. Please correct the ‘-2’ of ’10-2s’ in superscript form on line 174

3. Please correct ‘3d’ to 4d on line 248

4. What is the zeta potential of E1, E2 and E3?

5. Please confirm the structure of Zein in Figure 6b

6. There is lack of description of in vitro drug release method in Section 2. Materials and Methods. Please add the description. In addition, the UV spectra of pure CIP was shown in Figure 9a, however the other components in E3 may also interfere the UV spectra of CIP. Please add the UV spectra of E3 as supporting information. If the UV of CIP was interfered, please explain how to eliminate the interfere.

7. Only CIP release was detected; however, Yunnan Baiyao release was not detected. Please evaluate release profile of Yunnan Baiyao if possible.

8. Although the antibacterial activity was evaluated at certain concentration with different time point, the MIC of E3 was not provided. Please add MIC information of E3 if possible.

9. Back to the design rationale, please explain why Yunnan Baiyao was selected to formulate Zein particles other than CIP.

Moderate editing of English language required

Author Response

See attached file please!

Round 2

Reviewer 2 Report

It is acceptable now.

Reviewer 4 Report

The revised version addressed most of my concerns.